# Edge-Supervised Linear Object Skeletonization for High-Speed Camera

**DOI:** 10.3390/s23125721

**Published:** 2023-06-19

**Authors:** Taohan Wang, Yuji Yamakawa

**Affiliations:** 1Graduate School of Engineering, The University of Tokyo, Tokyo 113-8654, Japan; 2Interfaculty Initiative in Information Studies, The University of Tokyo, Tokyo 113-8654, Japan; y-ymkw@iis.u-tokyo.ac.jp

**Keywords:** high-speed, skeletonization, camera, binary image, linear object

## Abstract

This paper presents a high-speed skeletonization algorithm for detecting the skeletons of linear objects from their binary images. The primary objective of our research is to achieve rapid extraction of the skeletons from binary images while maintaining accuracy for high-speed cameras. The proposed algorithm uses edge supervision and a branch detector to efficiently search inside the object, avoiding unnecessary computation on irrelevant pixels outside the object. Additionally, our algorithm addresses the challenge of self-intersections in linear objects with a branch detection module, which detects existing intersections and initializes new searches on emerging branches when necessary. Experiments on various binary images, such as numbers, ropes, and iron wires, demonstrated the reliability, accuracy, and efficiency of our approach. We compared the performance of our method with existing skeletonization techniques, showing its superiority in terms of speed, especially for larger image sizes.

## 1. Introduction

Skeletonization is a widespread technique in the field of computer vision used to reduce an object’s dimension and represent it with a simplified “skeleton.” This approach conserves the object’s structure, topology, and properties, while significantly reducing the computational resources required. Consequently, this technique has found applications in diverse fields, such as character recognition, human gesture tracking, cable detection, etc.

As Figure 1d shows, conventional skeletonization approaches generally reduce two-dimensional or three-dimensional objects to curves that embody an object’s “medial axis”, which is commonly known as “thinning” [1]. In general, these algorithms can be grouped into two primary categories: classic mathematical techniques and deep learning-based methods.

Traditional skeletonization methods generally use the thinning process [1], whereby the image is iteratively eroded layer by layer from the object’s boundary. This process is based on Blum’s Medial Axis theory [2], which simulates a grass fire to uniformly burn away pixels from the boundary toward the interior.

Morphological thinning modified this “burning process” through constrained mathematical morphological erosion, while still incorporating the same “burning” concept. Other Voronoi diagram-based algorithms also use this idea of finding the skeleton by considering the object’s topology [3,4,5]. While the MAT method produces skeletons that are less susceptible to topological errors, compared to thinning-based methods, it can be computationally expensive, especially for complex objects [6].

In recent years, some contemporary researchers have suggested deep learning-based techniques for skeleton extraction. For instance, the DeepSkeleton, proposed by Wei Shen [7], is a fully convolutional network, which is designed to extract the skeleton in different scales from multi-stages. The Multi-Scale Bidirectional Fully Convolutional Network (MSB-FCN) [8] employs a bidirectional structure to capture multi-scale feature representations of deep features of the network to learn the information from multiple sub-regions.

In addition to using the fully convolutional network, some methods, such as that in [9,10,11,12], use U-Net-based networks to treat skeletonization as a semantic segmentation problem. Additionally, Liu et al. proposed the Rich Side Output Residual Network (RSRN) [13], which fuses Side-output Residual Network (SRN) for object symmetry detection and Richer Convolutional Features (RCF) to improve the detection. W. Tang et al. [14] proposed synchronizing convolutional neural networks (CNNs) and long short-term memory (LSTM) to improve the detection accuracy for Chinese character recognition. These deep-learning-based skeletonization algorithms have proven robust in complex environments [14,15,16].

While convolutional neural networks can produce impressive results in processing complex images, their computational complexities make them unsuitable for high-speed applications. In real-time implementations, such as vibration monitoring, robotic manipulation, and autonomous vehicles, high-speed skeletonization is essential to provide robust results and increased responsiveness.

This paper presents a novel skeleton detection framework that is designed for high-speed results for binary images containing linear objects, such as the object shown in Figure 1b. Our method can be ten times faster than some of the most commonly adopted methods, such as Zhang’s method or Lee’s method. With such high speed, the method provides possibilities for various applications, including real-time monitoring of string vibrations and tracking rapid motions of ropes using high-speed cameras.

A summary of our contributions follows:We propose a novel edge-supervised skeletonization approach, specifically designed for high-speed skeleton extraction that does not need to scrutinize every pixel in a binary image.We introduce a branch detector and an intersection center detector to enhance the quality of our skeletonization outcomes by identifying branches and intersection centers of the object for skeleton searching.We develop an innovative skeleton detection framework to facilitate high-speed applications for binary images.

The paper is structured as follows. Firstly, we review the related algorithms used in our work. We then describe our pipeline for edge-supervised center searching and an intersection detector for extracting linear object skeletons in two sections. In Section 5, we present a series of experiments to demonstrate the effectiveness of our approach and compare it with commonly used methods, such as Zhang’s and Lee’s methods. Finally, we summarize our work and propose future research directions.

## 2. Related Work

High-speed skeletonization offers numerous benefits in various applications, particularly in image processing and computer vision. Such faster skeletonization allows for real-time processing of images or videos, enabling quicker decision-making and improved responsiveness in time-sensitive applications. Moreover, high-speed algorithms require fewer computational resources and reduce overall processing time, which enables the handling of larger datasets and higher-resolution images in parallel computing.

This work aims to develop high-speed skeletonization solutions for the binary images of linear objects using traditional methods, which are faster than deep learning techniques. Skeletonization algorithms can be grouped into three major categories based on their principles and the underlying object representation [16]:

(1) Algorithms based on Voronoi diagrams or continuous geometric approaches. The following methods can be categorized in this category:“power crust”, proposed by Nina Amenta, “regular set model”, proposed by Jonathan W. Brandt, and the procedure for the generation all [3,17,18]. (2) Algorithms based on the principle of the continuous evolution of object boundary curves. The method proposed in [19,20,21], using different model-based curve propagations to implement Blum’s grass fire propagation procedure, can be categorized in this category. (3) Algorithms based on the principle of digital morphological erosion or location of singularities on a digital distance transform field. The algorithms proposed in [22,23,24,25], using different strategies to speed the process of digital skeletonization algorithms, can be categorized in this category.

Several researchers have proposed improved skeletonization algorithms to address the limitations of existing algorithms. Lu et al., [26], improved skeletonization by preserving important structures, while Zhang et al. [27,28], introduced fast parallel thinning algorithms. Lee et al. [29] presented the three-dimensional parallel thinning method using an octree data structure. Some researchers use the Gaussian mixture model to extract the center of the objects and use constraints to control the skeleton structures as desired to achieve real-time applications [30,31,32]. However, these methods may not be robust for linear objects and do not satisfy the speed requirements for high-speed applications.

The methods mentioned above require examining all pixels within the binary, leading to a significant waste of computational resources. In contrast, our approach does not necessitate scrutinizing every single pixel in a binary image containing the linear object. Additionally, we fully capitalize on the edge information to supervise our center searching process, accelerating our algorithm.

## 3. Edge-Supervised Skeletonization System

Our method shares the main idea of Blum’s Medial Axis theory, which is to recognize the centers of an object as its skeleton. The objective of our algorithm is to fit a set of piecewise linear curves to the target object, where the skeleton curve can be interpreted as a connection of midpoints of image pixels. Specifically, we focus on finding the centers of an object at high speed, since center points are effective in representing linear object skeletons.

As Figure 2 demonstrates, our algorithm includes two main modules: Center Searching and Intersection Detector. The Center Searching is focused on extracting the centers of the target object with no self-intersection. The Intersection Detector is designed to separate the self-intersection linear object into branches with no intersection for the Center Searching module.

### 3.1. Edge Supervised Center Searching

In our algorithm, we define points $V=\left[v_{1},\cdots ,v_{m}\right]\subset R^{2}$ as the vertices of the skeleton, which correspond to the centers of the target object.

We reformulated the problem from skeletonization to finding the centers of linear objects. Instead of using circles to determine the centers, we use a more direct method that involves intersecting a line with the object. As is shown in Figure 3, Step 4, the center of the two intersection points of the line and the two edges is also the center of the linear object. Thus, a center point $v_{m}$ can be represented by two edge points with $v_{m}=\frac{p_{edge1}+p_{edge2}}{2}$, where $p_{edge}$ are the yellow intersection pixels of the edge and the line, as is shown in Figure 3.

As Figure 3 demonstrates, the linear structure is very suitable for this kind of center searching because of its relatively constant width. By continuously drawing straight lines across the entirety of the linear deformable object, we can link the resulting center points *V* together to extract the skeleton efficiently without examining every pixel in the binary image.

To ensure the correct search direction, we divide the process into four steps. First, we select an initial point and then find the normal vector of the object’s edge, as shown in Figure 3 Step 1 and Step 2. The normal vector is defined by an initial edge point and the angle Θ. The edge of the object contains information on the object’s contour, and the normal vector of the object’s edge at a given initial edge point helps determine whether the current searching direction is correct. Then, we use the normal vector to define a line segment and check its intersection with the object’s edge. By finding the intersection points, we can locate the center point of the linear object, as shown in Figure 3 Step 4. By repeating this process, we can efficiently extract all the center points of the linear object without checking all the pixels of the binary image.

The calculation of the normal vector is similar to the edge extraction process in computer vision. In our algorithm, we use the Prewitt operator [33] to determine the direction of the normal vector at an initial search point. The kernel for the Prewitt operator can be expressed in the following format:(1)$$K_{x}=\left[\begin{array}{ccc} 1 & 0 & -1\\ 1 & 0 & -1\\ 1 & 0 & -1 \end{array}\right],K_{y}=\left[\begin{array}{ccc} 1 & 1 & 1\\ 0 & 0 & 0\\ -1 & -1 & -1 \end{array}\right]$$

Since the Prewitt operator uses two 3 × 3 kernels to convolve with the pixels in the original image, we define *I* as an original binary image to convolve with the Prewitt operator. To ensure computational efficiency, we do not implement the convolution throughout the whole image. Here, we select *A*, a 3 × 3 pixels block from *I*, to convolve with the Prewitt operator. The center of this pixel block is the selected edge point $p_{edge}$.
(2)$$G_{x}=K_{x}A,G_{y}=K_{y}A$$
where $G_{x}$ and $G_{y}$ represent the magnitude of the convolution results of *x* and *y* directions, respectively.
(3)$$G=\sqrt{{G_{x}}^{2}+{G_{y}}^{2}},\Theta =atan2\left(G_{y},G_{x}\right)$$
where *G* is the magnitude of the convolution results. Θ represents the angle of the normal vector at an edge point.

The accuracy of the Θ can be affected by the smoothness of the detected edge through the Prewitt operator, and its calculation, based only on local pixel values, may not always be stable. To ensure robustness, we sample the Θ under *N* different resolutions of the binary image. The sampling of the *A* on different resolutions of the binary image is a downsampling process. For example, every pixel in $A_{n}$ averages n×n pixel values into a single pixel. Thus, $A_{n}$ contains the contour information 3n × 3n of the original image.

The final Θ is the average sum of all $\Theta_{i}$ values:(4)$$\Theta_{i}=atan2\left(K_{y}A_{i},K_{x}A_{i}\right)$$
(5)$$\Theta =\frac{\sum_{i=1}^{N}\Theta_{i}}{N}$$

To ensure computational efficiency, the downsampling process is applied only locally every time the search reaches the edge.

Once the Θ is acquired, we are able to select the searching direction. Based on the Θ, we shift a bias angle $\Theta_{bias}$ to ensure the search moves in the direction we desire, as demonstrated in Figure 3, Step 3. Here, we define the searching direction as *V*. In the very first initializing process for a linear object, a positive and a negative value is selected to search both directions of the object. The $\Theta_{bias}$ is the parameter we use to control the searching speed in the direction *V* within the linear object. With smaller $\Theta_{bias}$, we acquire more dense *V* points to describe the skeleton at the cost of sacrificing time consumption.

Our proposed method enables high-speed searching of the center points in a binary image containing a linear object. The algorithm flow is described in Algorithm 1, which takes a binary image as input and outputs a sequence of center points *V*. In the initial step, the algorithm searches in two different directions inside the linear object using a positive and a negative bias angle until the end of the object is reached.

To update the searching direction, the algorithm confirms each pixel inside the target object along the direction of $\Theta_{bias}$ until it reaches the edge pixel (lines 11 and 21). Upon detecting the edge pixel, a new Θ is computed to update the searching direction. This process iterates until the end of the object is reached, with all the center points stored in sequence in *V*.

### 3.2. Intersection Detector

Utilizing the aforementioned method, we can efficiently detect a linear object’s skeleton when the object does not intersect with itself. To detect the skeleton in cases where self-intersection occurs, we introduce a new intersection detection module. The purpose of this module is to identify any existing intersections and initiate new searches on these additional branches.

#### 3.2.1. Coarse Branch Detector

The process of detecting self-intersections in the object’s skeleton is illustrated in Figure 4. After each center pixel is searched, a checking circle is initialized to monitor the object’s width changes, as shown in Figure 4a. By intersecting the checking circle with the object’s edges, we can estimate the object’s width. If there is any sudden change in the width, indicating a possible intersection, the branch detection procedure is initialized. The center of the checking circle is the center pixel $v_{m}\in V$, and we check all the pixels on the circle with radius *r*. The object’s width *w* is defined as the distance between intersection points p1 and p2, as shown in Figure 4a.
**Algorithm 1** Edge Supervised Skeletonization for Linear Object.**Input:** binary image *I***Output:** skeleton points *V* 1:**procedure** 2:    select an initialization edge point $p_{edge}$ 3:    compute the normal vector Θ at the initialization edge point 4:    initialize a new search with positive $\Theta_{bias}$ 5:    **for** no end detected **do** 6:        **if** current pixel is on edge **then** 7:           compute the normal vector Θ 8:           set the searching direction to $\Theta_{bias}$ 9:           save the center point into *V*  10:        **else**  11:           move to the next pixel along the searching direction $\Theta_{bias}$  12:        **end if**  13:    **end for**  14:    initialize a new search with negative $\Theta_{bias}$  15:    **for** no end detected **do**  16:        **if** current pixel is on edge **then**  17:           compute the normal vector Θ  18:           set the searching direction to $\Theta_{bias}$  19:           save the center point into *V*  20:        **else**  21:           move to the next pixel along the searching direction $\Theta_{bias}$  22:        **end if**  23:    **end for**  24:    **return** position of the centers *V*  25:**end procedure**

In the search process, our algorithm continually monitors the width changes until the increase of the width is greater than a given threshold λ, which is the maximum percentage width increase that we allow:(6)$$\frac{3w_{i}-\sum_{j=i-3}^{j=i-1}w_{j}}{\sum_{i=j-3}^{i=j-1}w_{j}}\geq \lambda$$

In practical applications, it is unnecessary to check the status of all pixels on the circles when the width is relatively constant. Therefore, to improve computational efficiency, we only need to determine if the sampled points are inside or outside the object. The crucial factor is to ensure that the number of sampled points is sufficient to capture any width changes that exceed the threshold. To achieve this, we sample *m* points uniformly along the circle. The minimum required number of points can be defined as following Equation (7). The detailed proof can be found in the Appendix A.
(7)$$m>\frac{2\pi \sqrt{4-{\left(1+\lambda \right)}^{2}}}{\lambda }$$

As demonstrated in Figure 4, there are three new branches that need to be searched. For each new branch, a new search can be initialized to calculate the center points. The skeleton can then be described as the collection of all these center points on each branch.

Before initiating a new branch, selecting a suitable initialization point is crucial for a successful search. As shown in Figure 4b,c, the algorithm uses a circle to intersect with the object in order to determine an appropriate initial pixel for each branch. The radius of the circle is $R_{c}$. This step is referred to as coarse intersection selection. The primary objective of this procedure is to detect all the branches and provide a suitable search direction. The following cost function is used to evaluate the circle:(8)$$C_{circle}=n\sqrt{\frac{\sum_{i=1}^{n}\left(B_{width}-\mu_{width}\right)}{n}}$$

The cost function aims to detect as many branches *n* as possible while ensuring that the widths of all branches $B_{width}$ are as similar as possible. The average width of all branches $B_{width}$ is denoted by $\mu_{width}$. As shown in Figure 4b, the blue circle is the only one that can identify all three branches with similar $B_{width}$, thus providing a good initialization point for each branch.

Once all the branches are detected and separated, the next step is to initialize the center point search method as mentioned in Section 3, shown in Figure 4c.

#### 3.2.2. Intersection Center Detector

The following method is actually an adjustment procedure to refine the center of the intersection detected in Figure 4b.

With the previous coarse detector, we could only acquire a rough center for the intersection. However, in some extreme cases, the center may even be located outside the target object. In the refinement branch detection procedure, the target is to detect a more accurate and proper center of the intersection to connect the center points from different branches.

In the first step of this detector, *M* beams are evenly distributed at 360 degrees from the center of the coarse detection result and spread toward the edge of the target object or the circle from the coarse detection step. As Figure 5 demonstrates, there are six beams that cannot reach the maximum radius. The red portion of the beams is the missing part.

In the next step, the red vectors are treated as the “forces” $f_{i}$. Applying all these “forces” onto the center point, results in the “sum force” *F* vector trying to push the center point toward the center position. The magnitude of this force describes the amount of shift required to move the center point to the right position.

The magnitude of the component forces:(9)$$∥f_{i}∥=∥R_{c}-l_{i}∥$$
where $l_{i}$ is the length of the red beam, as demonstrated in Figure 5a. The direction of $f_{i}$ is the *i*th beam’s direction.

Resultant Force:(10)$$F=\sum_{i=1}^{M}f_{i}$$

Then, we update the current center position to a new position by adding the “Resultant Force *F*”. In this procedure, the adjustment is iteratively implemented until either the magnitude of *F* is small enough within the tolerance or the maximum iterations is reached.

#### 3.2.3. Edge Supervised Searching with Branch Detection

With the intersection detector, our method is summarized in Algorithm 2. In the initialization step, we do not need to use positive and negative biases to start the search in two directions as before. Instead, we use the branch detection method to detect the branches that need to be searched through at the selected initialization edge point. The branch detection module automatically detects any branches that need to be searched. Differing from Algorithm 1, the branch detection procedure is included each time a center is detected. Whenever new branches are detected, they are added to the branch list $B_{list}$ waiting to be searched. With the intersection detector, we can detect all the branches and initialize center points searches on each of the branches.
**Algorithm 2** Edge Supervised Skeletonization for the Linear Object with Branch Detector.**Input:** binary image *I***Output:** skeleton points *V* 1:**procedure** 2:    select an initialization edge point $p_{edge}$ 3:    compute the normal vector $\Theta_{bias}$ at the initialization edge point 4:    initialize a branch detection and save branches in branch list $B_{list}$ 5:    **for** branches in $B_{list}$ **do** 6:        computes the normal vector at the branch initialization edge point 7:        initialize a new search with positive $\Theta_{bias}$ 8:        **for** no end detected **do** 9:           check the current pixel  10:           **if** current pixel is on edge **then**  11:               compute the normal vector Θ  12:               set the searching direction to $\Theta_{bias}$  13:               save the center point into *V*  14:               initialize a branch detection  15:               **if** new branches detected **then**  16:                   save new branches in branch list $B_{list}$  17:               **end if**  18:           **else**  19:               move to the next pixel along the searching direction $\Theta_{bias}$  20:           **end if**  21:        **end for**  22:    **end for**  23:    **return** position of the centers *V*  24:**end procedure**

## 4. Experiments and Results

In this section, we evaluated the proposed method using binary images of different objects, such as numbers, ropes, and wires. The efficiency of the method was tested on images of various sizes. To ensure efficiency, the algorithm was implemented in C++ with CMake and compared with Zhang’s method, Lee’s method, Medial axis skeletonization, and Morphological thinning. The algorithm was implemented in real-time on a high-speed ximea MQ013CG-ON camera. During the implementation, we set the diameter of the circle in the branch detection module to twice the width of the linear object calculated from the initialization step. The experiments demonstrated the speed and accuracy of the method on both simulation data and real data.

As shown in Figure 6, our algorithm efficiently found the skeletons of different binary image shapes. In the “number” images, the left part shows the original binary images and the right part shows the extracted skeleton represented by black lines. The extracted skeleton successfully represented the structure of the numbers. However, when the image had sharp corners, such as the numbers “5” and “7”, our algorithm could not find perfect center points to represent the skeleton. This was due to the limitation of the “edge-supervised search”. At the “corner” of the “7” image, there was a sudden change in the edge’s normal vector direction, which meant that the edge lost the supervised function at such positions. In some extreme cases, the supervised function might even be wrong. Currently, our solution is to select a new edge point to start a new center point search in order to skip the abnormal edge changes.

The right parts of Figure 6 show the results of testing the proposed method on images of ropes and iron wires. The left part displays the original images, and the green lines on the right demonstrate the detected skeletons. Compared to the number of images, the skeletons for ropes and iron wires were smoother and more accurate. This was because the edges of the ropes and iron wires were smoother and capable of providing more reliable direction supervision for searching. The results indicated that our algorithm could efficiently and accurately find a reasonable “medial axis” that represented the major structure of various types of linear objects, including numbers, ropes, and iron wires.

### 4.1. Experiments of Time Consumption

The primary goal of this research was to achieve high speed in extracting linear objects’ skeletons from binary images. To demonstrate the efficiency of our algorithm, we performed the following speed tests to demonstrate the time consumptions for different sizes of images. The results are compared with some of the most commonly used methods: Zhang’s method, Lee’s method, Medial axis skeletonization, and Morphological thinning.

As is shown in Table 1, our algorithm outperformed the other methods in terms of speed. This advantage was more pronounced with larger image sizes. The reason for this is that our algorithm focuses on searching through the interior of the target object, which is distinct from other methods. The time consumption was mainly influenced by the diameter of the target object and the number of iterations searched inside the target object. The reason that the time consumption increased as the size of the image increased was mostly because the diameter of the target objects also increased. However, if the size of the target object remained constant while the image size increased, the time consumption would remain relatively unchanged.

During the experiments, we also collected time consumption data for our algorithm without the intersection detector. As the data in the second and third columns in Table 1 and Figure 7b show, the intersection detector resulted in an extra 10% to 15% time consumption. However, both algorithms were suitable for high-speed data processing.

In the experiments, the computation time was significantly influenced by the shape and width of the target object. Changing the shape of the target object, even with the same image size, could result in varying time consumption. Therefore, the results in Table 1 should only be used as a general indication of the speed of our algorithm.

### 4.2. Experiments with Simulation Data

To evaluate the performance of our method, we used simulation data, as acquiring ground-truth data from a real-world object is hard. In Figure 8, we present the results of the simulation data. Firstly, we created the ground truth skeletons, as shown on the very right of Figure 8. Then, we increased the thickness of the skeletons to ensure that the ground truth was the exact center of different shapes, as shown in the image on the left. Finally, we compared the ground truth with the result of our simulation data.

As Figure 8 demonstrates, our algorithm could extract the skeleton correctly from all the binary images. The error of the data in Figure 9 indicated that the shape of Figure 8A had a slightly larger error than the other shapes. The reason for this error difference was that the cross-section interfered with the edge-supervised searching process. Despite using an intersection detection algorithm, the algorithm still lost edge supervision near intersections. However, the algorithm was able to extract the skeleton structure and accurately determine the cross-section center, preserving the topological structure of the original object. The average error for all the simulation data was approximately 0.3 pixels, demonstrating the effectiveness of our algorithm. The achieved sub-pixel accuracy demonstrated that our algorithm performed at the same level as other algorithms when applied to linear objects.

## 5. Conclusions and Future Works

In this paper, we propose a novel high-speed skeletonization algorithm that uses edge supervision and a branch detector to detect the skeleton of a linear object’s binary image. Our proposed method can be applied to applications such as rope, wire, and cable vibration monitoring and tracking. Moreover, the algorithm shows promising potential for parallel computing, which can further enhance its speed.

By focusing on the edges of the object, we were able to efficiently search inside the object, resulting in faster processing times. Furthermore, the branch detector allowed us to handle self-intersections with only an extra 10% to 15% time consumption, which also improved the accuracy of our algorithm.

We conducted a series of experiments to validate the reliability and speed of our algorithm. Our results showed that, for 512×512 binary images, our algorithm could output results at 1 kHz, demonstrating its high-speed processing capabilities.

Moving forward, we plan to improve our cross-section detection method to handle more complex shapes, such as Chinese characters. Additionally, we aim to improve the speed of our algorithm by optimizing the selection of pixel points in the search process. Finally, we plan to test our algorithm in other implementation cases, such as vibration observation and robot interactions, to further validate its effectiveness in real-world scenarios.

## Figures and Tables

**Figure 1 sensors-23-05721-f001:**
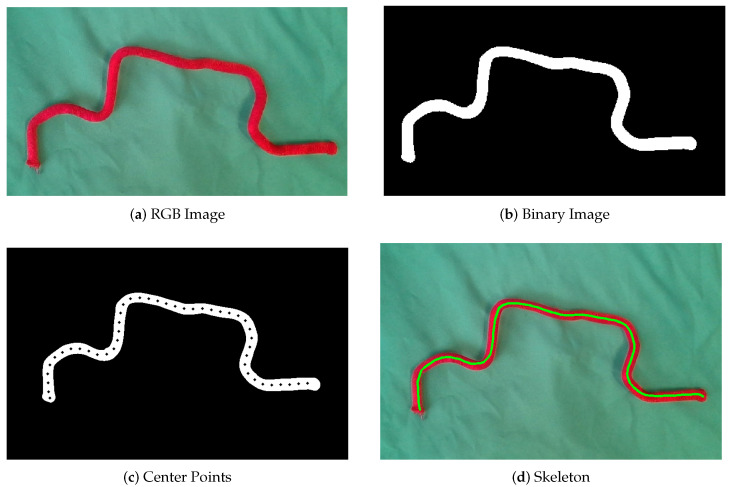
Demonstration of the process to extract skeleton from a depth image.

**Figure 2 sensors-23-05721-f002:**
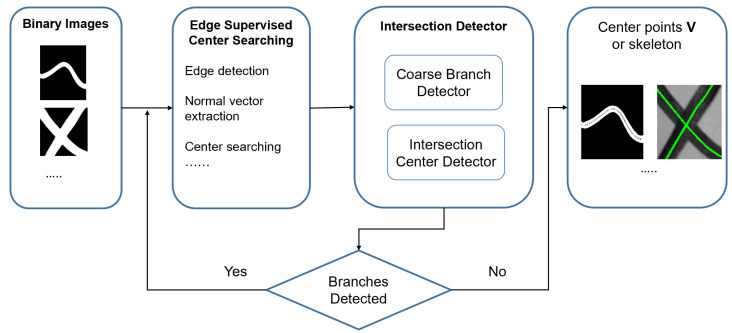
This figure shows the structure of our algorithm. The inputs are binary images and the outputs are points or skeletons. The two main modules of the system are the Edge Supervised Center Searching and Intersection Detector.

**Figure 3 sensors-23-05721-f003:**
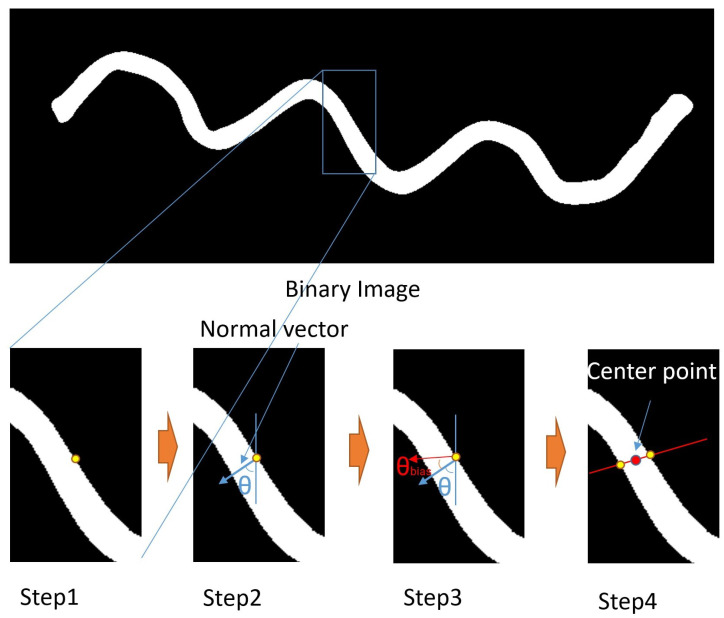
This figure describes the process of computing a center point of a linear object. The upper image is a binary input image. In the first step, an initial edge point is chosen. In step 2, the normal vector of the edge at this initial edge point is calculated. Step 3 adds a bias angle to give a searching direction for step 4. In step 4, the edge point on the other side is found. With these two edge points, the center point can be acquired.

**Figure 4 sensors-23-05721-f004:**
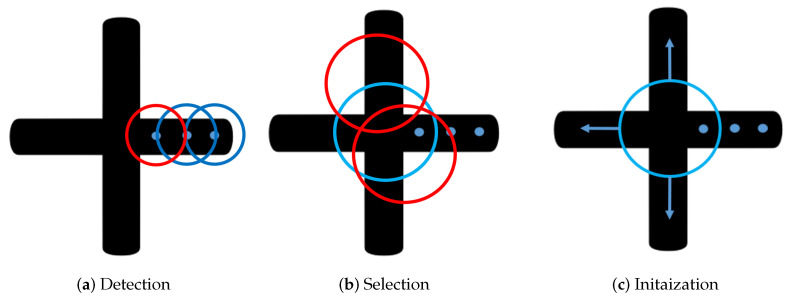
The figure shows the process of initializing the new branch detection. In the Detection step (**a**), the circle keeps monitoring the width change of the object. Once the intersection is detected (sudden width change detected, which is the red circle in (**a**)), the algorithm selects a circle able to find all the branches, as demonstrated in (**b**). As is shown in (**c**), with all the branches detected, new searches are initialized on each of the branches.

**Figure 5 sensors-23-05721-f005:**
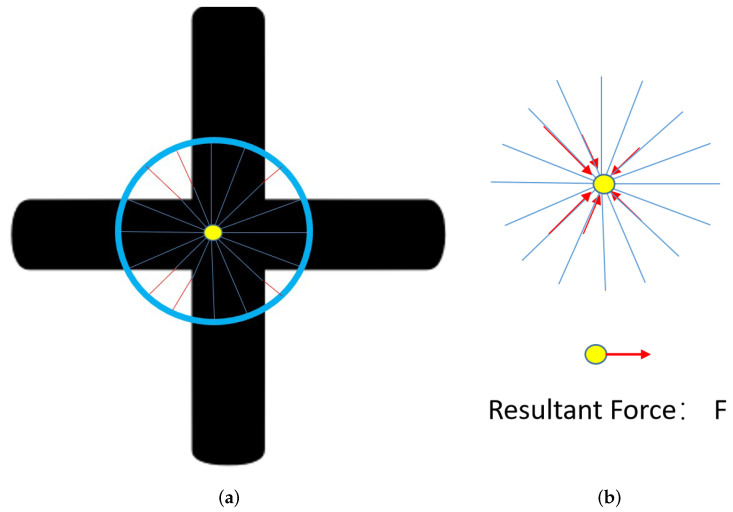
As is shown in (**a**), 16 beams are evenly distributed at 360 degrees from the center of the coarse detection circle. (**b**) shows components of the resultant force *F* are the red vectors from (**a**).

**Figure 6 sensors-23-05721-f006:**
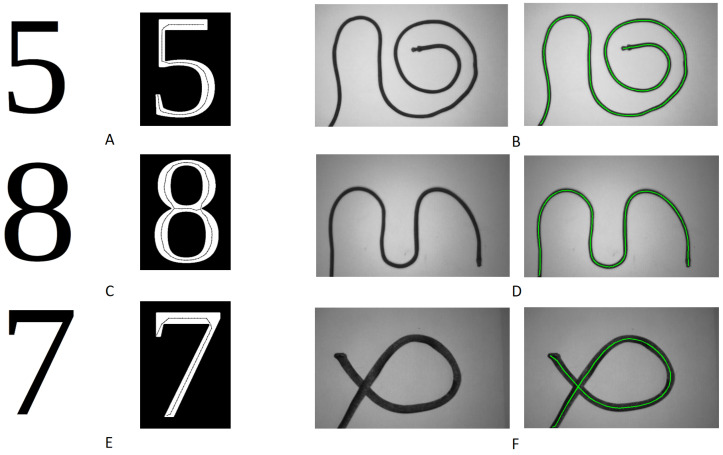
This is the demonstration of the skeletonization results on number images, cables, ropes, and wires. To apply our algorithm, we needed to transform the right parts of the grayscale images into binary images. The green lines are the detected results. The figures (**A**,**C**,**E**) are the detection results of number binary images. The figures (**B**,**D**,**F**) are the detection results of rope, iron wire, and cable.

**Figure 7 sensors-23-05721-f007:**
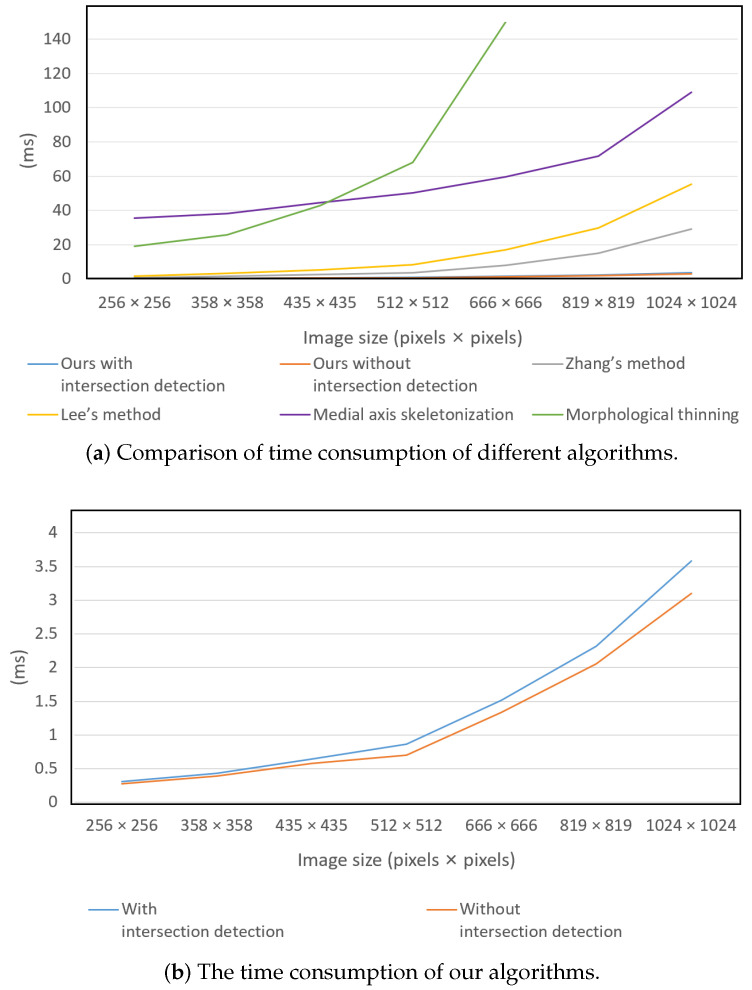
Demonstration of time consumption versus image sizes.

**Figure 8 sensors-23-05721-f008:**
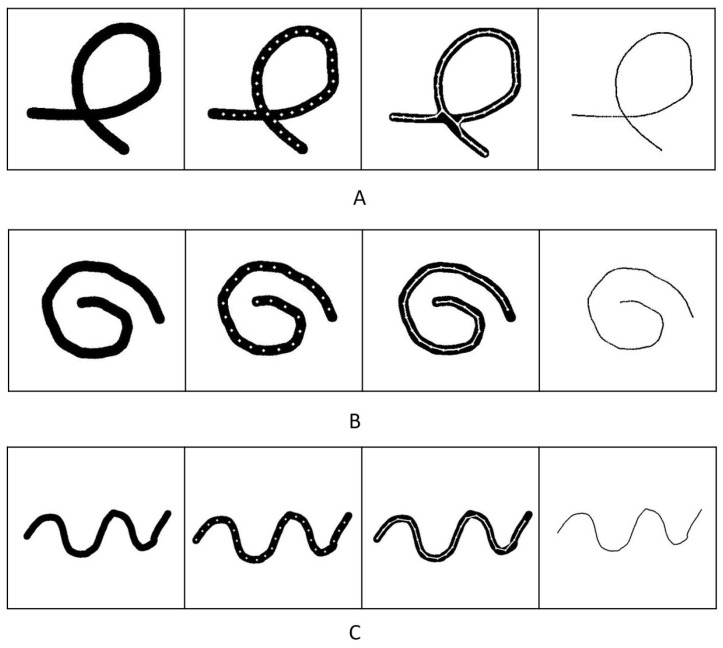
Demonstration of the skeleton detecting results of simulation data with different shapes. (**A**) demonstrates the result of a self-intersected object. (**B**,**C**) are the results of some randomly generated simulation objects.

**Figure 9 sensors-23-05721-f009:**
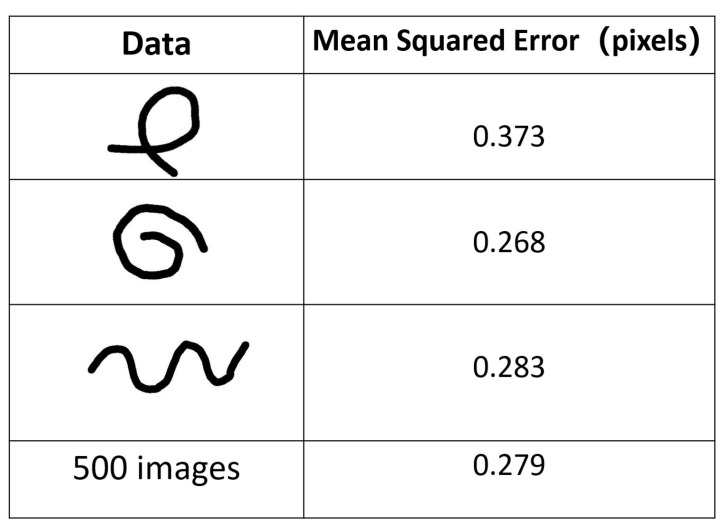
Mean Squared Error of the simulation data.

**Table 1 sensors-23-05721-t001:** The time consumption (ms) for different sizes of images with our algorithms and other state-of-the-art algorithms.

Image Size	Ours with Inter- Section Detector	Ours Without Intersection Detector	Zhang’s Method	Lee’s Method	Medial Axis Skeletonization	Morphological Thinning
256×256	0.31283	0.275	0.7511	1.544	35.565	19.120
358×358	0.43038	0.388	1.529	3.260	38.250	25.816
435×435	0.64367	0.581	2.565	5.332	44.685	42.815
512×512	0.86207	0.701	3.804	8.375	50.191	68.088
666×666	1.52092	1.334	8.137	17.174	59.705	150.337
819×819	2.31342	2.057	15.175	29.789	71.856	279.808
1024×1024	3.58309	3.102	29.126	55.387	109.202	431.654

This table demonstrates the processing speed of these algorithms on different image sizes. In different applications, the processing speed is also influenced by the shape and size of the object itself.

## Data Availability

The study did not report any data.

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
