# Peer review of "Edge-Supervised Linear Object Skeletonization for High-Speed Camera"

_sensors, 2023, doi:10.3390/s23125721_

Round 1
Reviewer 1 Report
The paper is of good quality, scientific sound. The contribution is well described and the methodology can be reproduced.
The authors should better describe equations 9 and 10. The result should be in two dimensions, and currently it seems it is a scalar.
While understandable from the context, the authors should mention what w means in equation 6.
There are slight inconsistencies in the presentation. In line 248, "number" images are mentioned, but they are not included in the paper.
Another modification that is necessary is to better align the figures with the place they are first mentioned in text. Now they can be pages away and it makes the reading slight difficult.
There are some figures for which the reference in text shows "??" These should be fixed too.
Author Response
Thank you for your valuable feedback on our paper. We have carefully considered your suggestions and have made the following revisions to address the concerns raised (Please see the attachment. The pdf is the revised version):
Equations 9 and 10: In the revised version of the paper, we have provided a more detailed explanation of these equations, clarifying that the results are in two dimensions rather than scalar values. (Equation 9 represents scale value. The direction of fi is the ith beam’s direction.)
Equation 6: We apologize for the oversight in not explicitly mentioning the meaning of "w" in Equation 6. In the revised manuscript, we have included a brief statement explaining that "w" represents the linear object’s width.
Inconsistencies in the presentation: The mentioned "number" images in line 248 are included in Figure 6 which seems not to be displayed in the manuscript. We fixed the Figure 6 missing issue in the revision.
Alignment of figures: In the revised version, we realigned the figures more closely with their respective discussions.
Figure references as "??" in text (line: 259 and 270 in the revised version): We apologize for any confusion caused by the missing Figure 6 references in the previous version. We have rectified this issue by ensuring that all figure references in the text are now correctly labeled and properly cited.
Once again, we sincerely appreciate your valuable suggestions and feedback, which have significantly contributed to the improvement of our paper. We believe that the revised manuscript now addresses the concerns you raised and provides a more comprehensive and accurate account of our work.

Reviewer 2 Report
This manuscript presents a high-speed algorithm for detecting the skeletons of linear objects from binary images. The algorithm efficiently extracts accurate skeletons by using edge supervision, a branch detector, and addressing self-intersections. Experimental results demonstrate its reliability, accuracy, and efficiency on various binary images. The method outperforms existing techniques, especially for larger images, in terms of speed. The author compares the performance of the method with existing skeletonization techniques, highlighting its superiority in terms of speed, particularly for larger images. In summary, the study is interesting and provides valuable results. However, the current document does have several weaknesses that should be addressed to enhance its value as a publication.
(1) Content-wise, the document is acceptable. However, it is suggested that the author strengthens proofreading to avoid common mistakes such as incorrect typesetting, repetitive use of the same words and expressions, incorrect use of punctuation rules, and absence of spaces between words.
(2) Additionally, the document contains 25 references, with only 6 published in the last five years (24%), 1 published in the last five to ten years (4%), and 18 published over ten years ago (72%). Therefore, the total number of recent references is insufficient.
(3) Chapter 1, Introduction: Each paragraph in the introduction section is excessively long, and the logical flow lacks smoothness. It solely lists examples of deep learning in the visual field without logical organization. It is recommended to improve the organization of this chapter by summarizing the specific data that highlights the improvements of the proposed algorithm compared to other existing algorithms.
(4) The improvements of the algorithm proposed in this article compared to other algorithms can be succinctly summarized using specific data in this chapter.
(5) The authors may add more state-of-art CV application articles for the integrity of the manuscript (Detection and counting of banana bunches by integrating deep learning and classic image-processing algorithms; Computers and Electronics in Agriculture. Optimization strategies of fruit detection to overcome the challenge of unstructured background in field orchard environment: a review; Precision Agriculture.).
(6) Chapter 3: Edge-Supervised Skeletonization System: The algorithms discussed in this chapter employ traditional algorithms to quickly find the center point as the skeleton. However, they are primarily based on simple linear targets without complex backgrounds. It is important to evaluate the effectiveness of this algorithm in complex environments.
(7) Chapter 4: Experiments and Results: How does this algorithm overcome sudden changes in the direction of the normal vector at the edge?
(8) There is a textual issue with the mention of "Figure ??" in line 247.
(9) The colors of the curves "Ours with intersection detection" and "Medical axis sketchization" in Figure 7(a) are too similar, and the scale of the graph is inappropriate. The intervals between 0-50ms are too dense.
(10) Chapter 5: Conclusions and Future Works: The article has a limited number of references, and many of them are from ten years ago.
(11) The algorithms in this article utilize traditional algorithms to quickly find the center point as the skeleton, but they are primarily based on simple linear targets without complex backgrounds. Have you considered testing the performance of these algorithms in complex backgrounds instead of solely relying on simple "5" and "7" images?
(12) The algorithm presented in this paper is a fast skeletonization algorithm based on traditional methods. It would be beneficial to discuss the advantages and disadvantages of skeletonization algorithms based on deep learning techniques.
Author Response
Thank you for your thorough evaluation of our manuscript. We have carefully reviewed your suggestions and have made the following revisions (Please see the attachment. The pdf is the revised version):
(1)Proofreading and language improvements: We have conducted proofreading and language editing to rectify these issues and ensure a more polished and error-free manuscript.
(2)Insufficient recent references: We acknowledge the concern regarding the number of recent references in the document. In response, we have expanded our literature review to include more recent publications. The revised manuscript now includes 8 more recent papers to balance the references.
(3)Chapter 1, Introduction: We appreciate your feedback regarding the organization and logical flow of this chapter. In the revised version, we have restructured the introduction, providing a more coherent and logical presentation of the proposed algorithm's improvements over existing methods. Also, we have highlighted our speed improvements against some of the most commonly used methods.
(4)In the revised version, we have mentioned that our method can be 10 times faster than some of the most commonly adopted methods, such as Zhang’s method or Lee’s method.
(5)Addition of state-of-the-art CV application articles: We have taken your suggestion into account and included additional references to state-of-the-art computer vision application articles. These additions contribute to the integrity and relevance of the manuscript, further enriching the discussion.
(6)Evaluation of the algorithm in complex environments: We understand the importance of evaluating the algorithm's effectiveness in complex backgrounds. To address this concern, we have conducted some experiments in Chapter 4, such as the results demonstrated in Figure 6.
(7)Overcoming sudden changes in the direction of the normal vector: We've added a brief illustration to the solution: Currently, our solution is to select a new edge point to start a new center point search in order to skip the abnormal edge changes. Since this solution is very engineering, we think it may be better to not include all the details.
(8)Textual issue with "Figure ??" in line 247 (line: 259 and 270 in the revised version): We apologize for the oversight. The issue has been resolved, and the correct Figure 6 reference is now provided in the revised manuscript.
(9)Improvements to Figure 7(a): We appreciate your observation regarding the colors and scale of the graph. In response, we have adjusted the colors of the curves and optimized the scale to ensure better visibility and readability.
(10)Limited number of references in Chapter 5: We have taken your feedback into consideration and have expanded the number of references. The revised version now includes a broader range of references, encompassing recent publications in the field.
(11)Testing performance in complex backgrounds: We apologize for the missing Figure 6. Other experiments are demonstrated in Figure 6. In fact, we have conducted plenty of experiments on different objects, such as rope, iron, and wires.
(12) We have included some brief illustrations of the advantages and disadvantages of skeletonization algorithms based on deep learning techniques in the introduction part of the revised version.
We sincerely appreciate your valuable feedback and suggestions, which have significantly contributed to improving the manuscript. We believe that the revised version now addresses the weaknesses mentioned and provides an enhanced and more comprehensive account of our work.

Reviewer 3 Report
In general, the manuscript is written in a good English. It is easy to read and follow. The description of the proposed method is good and suitable for a scientific publication. (Remark: At some places, Latex references are incorrectly compiled and ?'s are present. Please correct them.) However, the novelty of the proposed method remains unclear. The authors mainly use traditional image processing techniques. In the introduction section, the authors should declare the contributions. A separate contributions subsection would be welcomed. Moreover, the possible applications of the proposed method is also unclear. The illustration of several applications would be welcomed.
Author Response
We would like to express our gratitude for your positive comments on the language and readability of our manuscript, as well as your acknowledgment of the suitability of the proposed method for scientific publication. We have carefully considered your suggestions and have made the following revisions to address the concerns raised(Please see the attachment. The pdf is the revised version):
Figure references as "??" (line: 259 and 270 in the revised version)in text: We apologize for any confusion caused by the missing Figure 6 references in the previous version. We have rectified this issue by ensuring that all figure references in the text are now correctly labeled and properly cited.
The novelty of the proposed method: We understand your concern regarding the clarity of the novelty in our work. To address this, we have revised the introduction section to explicitly declare the contributions of our research. In addition, we have included a separate part "A summary of our contributions" to clearly outline and emphasize the novel aspects of our proposed method.
Applications of the proposed method: We acknowledge that the possible applications of our proposed method were not adequately illustrated in the initial version. In the revised version, we briefly discussed the potential application of our method at the beginning part of the conclusion.
Once again, we sincerely appreciate your valuable feedback, which has greatly contributed to enhancing the quality and clarity of our manuscript. We believe that the revised version now effectively addresses the concerns you raised and provides a more comprehensive account of our work.

Round 2
Reviewer 3 Report
Contributions are still not clear to me.
1. Contribution: We have proposed a novel edge-supervised skeletonization approach, specifically designed for high-speed skeleton extraction.
Q.: What is the novelty in this edge-supervised skeletonization approach? Why is it suitable for high-speed skeleton extraction?
2. Contribution: We have introduced a branch detector and an intersection center detector to enhance the quality of our skeletonization outcomes.
Q.: What is the novelty of the branch detector compared to other solution in the literature? Why does it enhance the quality?
3. Contribution: We have developed an innovative skeleton detection framework to facilitate high-speed applications for binary images.
Q.: For example? Could you mention several applications?
Author Response
Thank you for your valuable feedback. (Please see the attachment. The pdf file is the revised version.)
Contribution1
Response: The novelty of our edge-supervised skeletonization approach lies in its combination of edge information and supervision to optimize the search process. By leveraging the distinct features of the object edges, our approach reduces unnecessary computations and focuses on searching within the object boundaries. This enables efficient skeleton extraction, making it suitable for high-speed applications. We have emphasized the novelty briefly in our revised version by adding: the method does not need to scrutinize every single pixel in a binary image. More details can be found at the end of the related works part.
Contribution2
Response: The novelty of our branch detector lies in its ability to detect branches and provide a reasonable center for the edge-supervised center searching process. Because our method is very different from most of the skeletonization approaches, this branch detection and intersection center searching method is specifically designed for this specific task. Thus it's hard to evaluate and compare with other skeletonization methods separately. The branch detector improves the quality of the skeletonization outcomes by preserving the connectivity and topology of the object.
Regarding the intersection center detector, its novelty lies in its capability to detect and initialize new searches on intersecting branches. This module ensures that self-intersections are appropriately handled, preventing errors in the skeletonization process and improving the overall quality of the results.
In the revised version, we added a simple illustration: enhance the quality of our skeletonization outcomes by identifying branches and intersection centers for skeleton searching.
Contribution3
Response: In the revised version, we added the examples before the summary of the contribution: the method provides possibilities for various applications, including real-time monitoring of string vibrations and tracking rapid motions of ropes using high-speed cameras. In fact, this algorithm is designed for the project of tracking iron wire and rope vibration when human interaction is involved with a high-speed camera at 500hz.
We sincerely appreciate your insightful comments and suggestions.
